# Bioinformatics-based prediction of conformational epitopes for human parechovirus

Hao Rong[1,2], Liping Wang[1,2], Liuying Gao[1,3], Yulu Fang[1,2], Qin Chen[4], Jianli Hu[1], Meng Ye[1,2], Qi Liao[1,2], Lina Zhang[1,2], Changzheng Dong[1,2]*

1 The Affiliated Hospital of Medical School, Ningbo University, Ningbo, China, 2 Department of Preventive Medicine, Zhejiang Provincial Key Laboratory of Pathological and Physiological Technology, School of Medicine, Ningbo University, Ningbo, China, 3 The Affiliated People's Hospital of Ningbo University, Ningbo, China, 4 HuaMei Hospital, University of Chinese Academy of Sciences, Ningbo, China

* dongchangzheng@nbu.edu.cn

**Data Availability Statement:** All relevant data are within the manuscript and its Supporting Information files.

**Funding:** This work was supported by the Zhejiang Fundamental, Public-interest Research Project

## Abstract

Human parechoviruses (HPeVs) are human pathogens that usually cause diseases ranging from rash to neonatal sepsis in young children. HPeV1 and HPeV3 are the most frequently reported genotypes and their three-dimensional structures have been determined. However, there is a lack of systematic research on the antigenic epitopes of HPeVs, which are useful for understanding virus-receptor interactions, developing antiviral agents or molecular diagnostic tools, and monitoring antigenic evolution. Thus, we systematically predicted and compared the conformational epitopes of HPeV1 and HPeV3 using bioinformatics methods in the study. The results showed that both epitopes clustered into three sites (sites 1, 2 and 3). Site 1 was located on the "northern rim" near the fivefold vertex; site 2 was on the "puff"; and site 3 was divided into two parts, of which one was located on the "knob" and the other was close to the threefold vertex. The predicted epitopes highly overlapped with the reported antigenic epitopes, which indicated that the prediction results were accurate. Although the distribution positions of the epitopes of HPeV1 and HPeV3 were highly consistent, the residues varied largely and determined the genotypes. Three amino acid residues, VP3-91N, -92H and VP0-257S, were the key residues for monoclonal antibody (mAb) AM28 binding to HPeV1 and were also of great significance in distinguishing HPeV1 and HPeV3. We also found that two residues, VP1-85N and -87D, might affect the capability of mAb AT12-015 to bind to HPeV3.

## Introduction

Human parechoviruses (HPeVs) belong to the species *Parechovirus* type-A of the *Picornaviridae* family. To date, 19 different HPeV genotypes (HPeV1-19) have been identified and HPeV1 and HPeV3 are the most prevalent types [1]. The extremely high seroprevalence of HPeV1 and HPeV3 (about 45–100%) indicates that HPeV1 and HPeV3 infections are common in young children. Although HPeVs infections usually cause mild diseases, including

(LGF18C060001), the Ningbo Natural Science Foundation (2018A610240), Medical and Health Program of Zhejiang Province (2019KY589) and the K.C. Wong Magna Fund in Ningbo University. The funders had no role in study design, data collection and analysis, decision to publish, or preparation of the manuscript.

**Competing interests:** The authors have declared that no competing interests exist.

gastrointestinal and respiratory diseases, rash, enteritis and diarrhea [2, 3], more than 60% of HPeV3-infected patients who were hospitalized had severe central nervous system diseases, including neonatal sepsis and meningitis [4–6]. There are no antivirals or vaccines available to combat HPeV infection except a few monoclonal antibodies (mAbs) may had the therapeutic potential [7, 8].

HPeV is a nonenveloped virus with an icosahedral symmetrical spherical structure and contains an ~7.3 kb single-stranded positive-sense RNA genome [9]. The polyprotein encoded by the HPeV genome is cleaved into three viral proteins (VPs) including VP1, VP0 and VP3 and seven nonstructural proteins (2A-C and 3A-D) [7]. In contrast to most picornaviruses such as foot-and-mouth disease virus (FMDV) and poliovirus I (PV1), HPeVs lack the maturation cleavage of VP0 into VP4 and VP2. The VPs reside in the HPeV capsid and assemble an asymmetric unit. Sixty units form the virus capsid [7]. The three-dimensional (3D) structures of HPeV1 [10] and HPeV3 [7, 9] have been determined at atomic resolution. Like other picornaviruses, the core secondary structures of HPeVs are made up of eight antiparallel β-sheets (βB-βI), which are further folded into β-barrels. There is a "canyon" on the surface of the virus capsid, and the "northern rim" of the canyon is predominantly formed by the VP1 BC and HI loops (Fig 1). The "southern rim" is formed by the "puff" and the "knob". The former consists of the VP0 EF loop and VP1 GH loop, and the latter is a short loop preceding the VP3 βB [11]. The surfaces of the HPeV virions are relatively flatter and shallower than that of FMDV and PV1 because the northern and southern rims of the canyon are less protruding in HPeVs [10].

Antigenic epitopes are useful for understanding virus-receptor interactions and finding potential cures for antiviral agents or molecular diagnostic tools [7, 8]. In addition, the determination of epitopes is of great significance to monitor the antigenic evolution of viruses [12–16]. Domanska *et al.* [7] discovered a human neutralizing mAb AT12-015 that specifically bound to HPeV3's flat region between the northern rim and knob and blocked attachment of virions to host cells. As for another genotype HPeV1, Shakeel [8] found that mAb AM28 could bind to the conformational epitope near the threefold vertex of HPeV1, which might inhibit HPeV1 uncoating by stabilizing the capsid. MAb AM18 could bind to the linear epitope of RGD motif in HPeV1 VP1 C-terminus, which blocked the binding of the integrin receptor [8]. There is, however, a lack of systematic research on the epitopes especially for conformational epitopes of HPeV1 and HPeV3, since the determination of the conformational epitopes requires complex techniques such as cryo-electron microscopy.

Previously, we developed a bioinformatics algorithm for human enterovirus based on Borley *et al.*'s multiple chains algorithm. The algorithm had successfully predicted the conformational epitopes for coxsackievirus A10 (CVA10) and found that the epitopes clustered into three sites [17]. In this study, we systematically predicted the conformational epitopes for HPeV1 and HPeV3 using similar method and compared them to discover the key residues for mAbs' binding.

## Materials and methods

### Sequence and structure analysis

The 3D structures of HPeV1 (Harris strain) [10] and HPeV3 (A308/99 strain) [7] with the respective PDB IDs of 4Z92 and 6GV4, whose resolutions were 3.10 Å and 2.80 Å, respectively, were downloaded from the RCSB PDB database [18]. The corresponding GenBank IDs of the amino acid sequences of HPeV1 and HPeV3, which were retrieved from the NCBI Nucleotide database, are L02971 and AB084913, respectively [19]. In addition, the complete genome sequences of HPeV1-19 strains were downloaded from the NCBI Nucleotide database to analyze the sequence conservation and the amino acid sequences of VPs were deduced. The

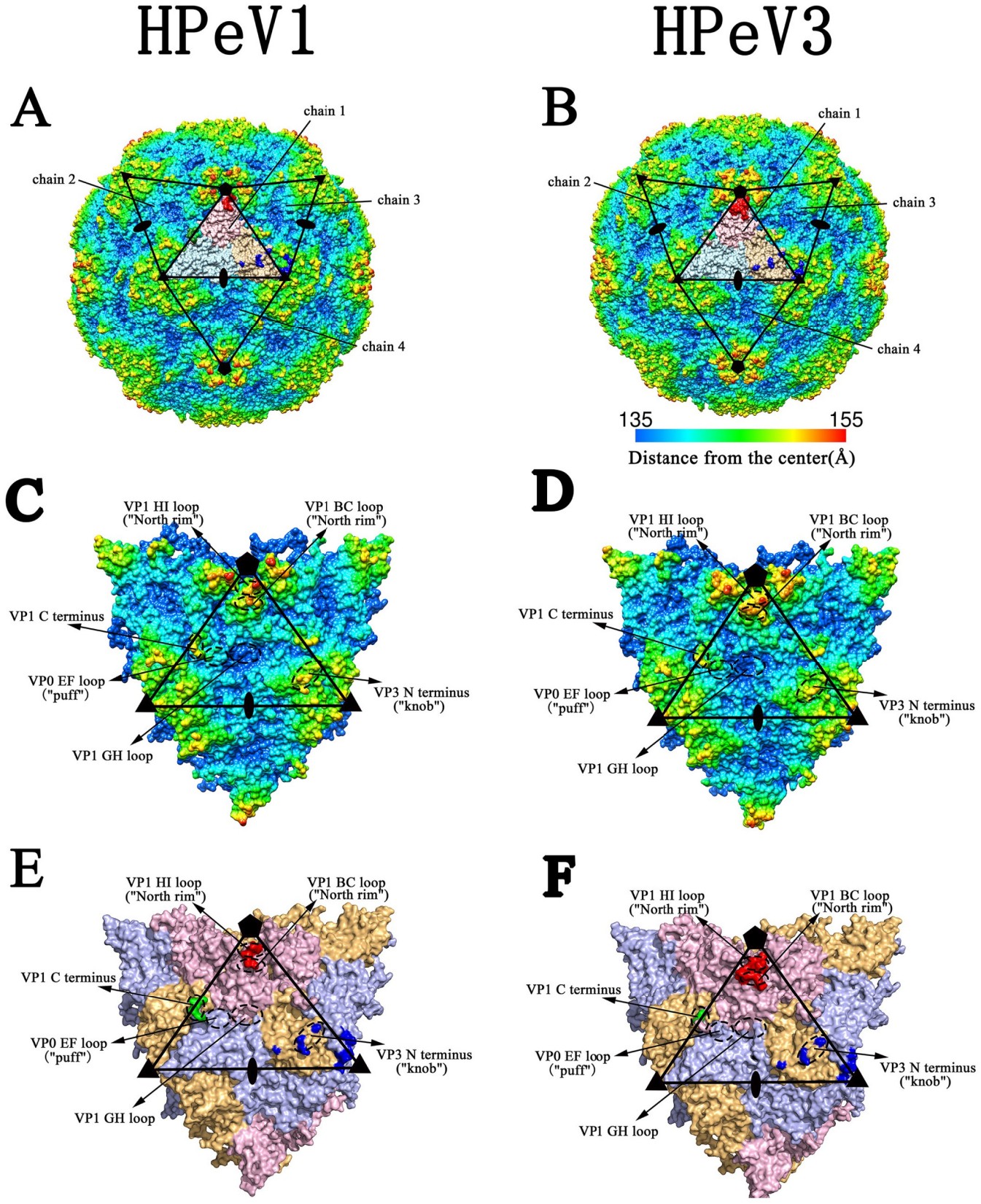

**Fig 1. The capsid structures of HPeV1 and HPeV3.** (A-B) Radius-colored surface representation of HPeV1 (A) and HPeV3 (B). The surfaces are colored blue to red, according to the distance from the particle center (red representing the furthest distance). The multiple chains, consisting of four subunits and labeled chains 1–4, are outlined in black. (C-D) Magnified views of the multiple chains and the capsid characteristics of HPeV1 (C) and HPeV3 (D) are shown. Chain 1 is indicated by a black triangle. (E-F) The predicted conformational epitopes of chain 1 of HPeV1 (E) and HPeV3 (F). Chain 1 is also indicated by a black triangle, with subunits VP1, VP0 and VP3 in light pink, light blue and light orange, respectively. Residues involved in conformational epitope sites 1, 2 and 3 of chain 1 are shown in red, green and blue, respectively. The positions of the icosahedral symmetry axes are indicated by pentagons (5-fold), triangles (3-fold), and ovals (2-fold).

strains used were as follows: HPeV1 (57 strains), HPeV2 (3 strains), HPeV3 (130 strains), HPeV4 (12 strains), HPeV5 (9 strains), HPeV6 (7 strains), HPeV7 (1 strain), HPeV8 (3 strains), HPeV14 (5 strains), HPeV17 (6 strains), HPeV18 (1 strain), and HPeV19 (1 strain). The amino acid sequences were aligned by MEGA 7 [20] using the MUSCLE tool. The sequence alignment results and the corresponding PDB files were uploaded to the online tool ESPript [21], which annotated the secondary structural information to the alignment results (S1 Fig). The figures for the 3D structures and surface characteristics of the viral capsids were generated using PyMOL [22] and UCSF Chimera [23]. The overall structural similarity measured by root mean square deviation (RMSD) was calculated using PyMOL [22]. All parameters used were the default values.

## Bioinformatics-based prediction of conformational epitopes

Borley *et al.* [24] developed a bioinformatics-based algorithm to predict the conformational epitopes for foot-and-mouth disease virus (FMDV) by generating multiple chains. HPeV and FMDV, both belonging to the *Picornaviridae* family, have analogous capsid surface structures. Therefore, we developed a bioinformatics algorithm based on Borley *et al.*'s research to predict the conformational epitopes for HPeV, which mainly included the following steps.

First, the complete capsids of HPeV1 and HPeV3 were generated based on 3D structure coordinates and noncrystallographic symmetry information in the PDB files. Afterwards, the central chain (chain 1) and surrounding chains (chains 2, 3 and 4) were selected to form multiple chains, and each chain consisted of the VPs VP0, VP1 and VP3 (Fig 1A–1D). The 3D structure of some residues on the VP1 C-terminus, including the RGD motif, were not determined in HPeV1. Therefore, we built a homology model for those residues to estimate the location of the RGD motif before generating multiple chains. Ultimately, three residues (VP1-217S, -218S and -219R) in HPeV1 were modeled using SWISS-MODEL [25], based on the 3D structure of the VP1 C-terminus of HPeV3.

Second, three classical and freely accessible conformational epitope online prediction tools were employed to predict the epitopes for multiple chains. These were Ellipro [26], DiscoTope [27] and Epitopia [28], of which the thresholds were 0.174, 0.3 and -10.7, respectively. The residues with predicted values above the thresholds were predicted as the candidate epitopes.

Third, the prediction results for chain 1 were selected. Then, the residues exposed on the surface of the viral capsid were selected. Briefly, we calculated the average distance between the Cα atoms of all residues on the viral capsid and capsid center. Any residue on the capsid with a distance that was greater than average was included.

Finally, the voting method was applied to determine the final epitopes, including the core and surrounding epitopes (S2 Fig). The residues, simultaneously predicted as candidate epitopes by three prediction tools (Ellipro, DiscoTope and Epitopia), were defined as the core epitopes. The residues (at the same location in the sequence alignment between two serotypes) were defined as the surrounding epitopes of HPeV3 when they were predicted to be candidate epitopes by any two prediction tools and were the core epitopes in other serotypes of the same species (such as HPeV1). Based on the assumption of "similar structure, similar epitope", we

had expanded the predicted epitopes by adding surrounding epitopes, which was the main point of our modification to Borley *et al.*'s algorithm.

## The accuracy of the predicted conformational epitopes

Borley *et al.* previously predicted the conformational epitopes of FMDV and compared them with known epitopes. We also predicted the epitopes of FMDV and compared them with Borley *et al.*'s results and known epitopes [29]. The PDB files for O1K and reduced O1K were not publicly available, thus, another two stains (O and SAT-2) were used instead. The PDB IDs for O, A, C, SAT-1 and SAT-2 are 1FOD, 1ZBE, 1FMD, 2WZR and 5ACA, respectively.

The antibody binding regions of HPeV1 and HPeV3, which were experimentally confirmed (experimental epitopes), were obtained by a literature search. The "footprint" of the capsid surface was generated using RIVEM [30]. Eventually, we evaluated the accuracy of the prediction algorithm by comparing the experimental epitopes with the predicted epitopes and analyzed the relationship between antibodies and conformational epitopes based on the footprint.

## Results

### Prediction results of the conformational epitopes of HPeV1 and HPeV3

The predicted conformational epitopes of HPeV1 and HPeV3 are shown in Table 1 and Fig 1. Similar to CVA10 [17], the conformational epitopes of HPeV1 and HPeV3 also clustered into three sites (sites 1, 2 and 3). Site 1 (shown in red in Fig 1E and 1F), consisting of the VP1 BC and HI loops, was located on the northern rim near the fivefold vertex. Site 2 (shown in green in Fig 1E and 1F), composed of the VP1 C-terminus, was located near the puff. The VP1 C-terminus of HPeV1, protruding from the viral capsid surface, contained an RGD motif binding to the αvβ6 and αvβ3 receptors [31]. The prediction results showed that conformational epitopes existed in this region. However, lacking the RGD motif, the VP1 C-terminus of HPeV3 can also form a epitope. Site 3 (shown in blue in Fig 1E and 1F) was divided into two parts: one was located on the knob and mainly composed of the VP3 N-terminus; the other was close to the threefold vertex and mainly consisted of the VP0 HI and VP3 HI loops. Compared to HPeV3, three amino acid residues were missing at the VP3 N-terminus of HPeV1 (S1 Fig); therefore, VP3-87T, -91N, and -92H in HPeV1 and VP3-90T, -94S, and -95S in HPeV3 were

**Table 1. Prediction results of conformational epitopes of HPeV1 and HPeV3.**

| Name of epitopes | Capsid characteristics | Secondary structure | HPeV1[a] | HPeV3[a] |
|---|---|---|---|---|
| Site 1 | North rim | VP1 BC loop | 84T, **85N** | 82D, 84N, **85N**, 87D |
| | | VP1 HI loop | **189G, 190T**, 191S, 192T | 187M, 188H, **189G, 190T**, 191T, 192R |
| Site 2 | Puff | VP1 C-terminus | 216T, 217S, **218S**, 219R[b] | 217G, **218S** |
| Site 3 | Knob | VP3 N-terminus[c] | **87T**, 91N, 92H | **90T**, 94S, 95S |
| | | VP3 EF loop[c] | 167S | 170T |
| | Threefold vertex | VP0 BC loop | 128P | - |
| | | VP0 HI loop | **254P, 255T, 256G**, 257S, **259N** | **254P, 255T, 256G**, 257A, **259N** |
| | | VP3 HI loop[c] | **221N, 222S, 223S** | **224N, 225S, 226S** |

[a]The bold letters represent the conserved amino acids between HPeV1 and HPeV3.

[b]Since some residues of the VP1 C-terminus in HPeV1 were not determined in the 3D structure, we generated the homology model for those residues using SWISS-MODEL.

[c]Compared to HPeV3, there are three amino acid residues missing in the N-terminus of HPeV1 VP3; therefore, VP3-87T in HPeV1 and VP3-90T in HPeV3 are actually at the same position in the sequence alignment.

actually at the same positions in the sequence alignment (S1 Fig and Table 1). In addition, the VP3 EF loop of HPeV1 and HPeV3 (167S and 170T, respectively) formed site 3 near the knob; the VP0 BC loop of HPeV1 (128P) participated in site 3 near the threefold vertex. It should be noted that Fig 1E and 1F shows the epitope distribution pattern, in which a few residues were from adjacent asymmetric units. The complete epitope distribution on the surface of the viral capsid is shown in S3 Fig, and we can see that site 2 on one asymmetric unit was actually very close to site 3 on the adjacent unit.

## The accuracy of predicted conformational epitopes

First, our algorithm was employed to predict the conformational epitopes of FMDV and the results were compared with Borley *et al.*'s (S1 Table). Because we had added surrounding epitopes to the prediction results through algorithm modification, the results should include Borley *et al.*'s epitopes. As expected, serotypes O, A, C and SAT-1 have 2, 3, 2 and 2 more epitopes than Borley *et al.*'s, respectively (S1 Table). However, serotypes C and SAT-1 had one less epitope on the VP2 EF loop. This was mainly due to the update of DiscoTope, which made the results slightly different from the previous. On the other hand, the added surrounding epitopes were not completely novel epitopes for FMDV. They were actually the epitopes of other serotypes. For example, there were epitopes predicted on the VP1 BC loop of serotypes A and C through algorithm modification. And these epitopes already existed in serotypes O, SAT-1 and SAT-2. Furthermore, the prediction results of surrounding epitopes were added through the assumption "similar structure, similar epitope", which made the distribution patterns of epitopes clearer (S2 Fig), and the differences in serotypes were mainly caused by the variations of amino acid residues on the epitopes. The prediction results of HPeVs can also observe such characteristics (Fig 3 and S1 Fig). These results proved that our algorithm modification was particularly suitable for epitope prediction and comparison of multiple closely related serotypes while ensuring the accuracy of prediction.

Secondly, the reported experimental epitopes of HPeV1 [8] and HPeV3 [7], obtained through a literature search, were identified by mAbs AM28 and AT12-015, respectively. Additionally, the RGD motif on VP1 of HPeV1 was a linear epitope binding to mAb AM18 [8]. The epitopes proposed above highly overlapped with our predicted epitopes, indicating that the prediction results were very accurate (Fig 2).

MAb AM28 bound to the residues on the VP0 BC, CD, and HI loops and the VP3 N-terminus, BC loop and EF loop of HPeV1. These binding regions highly overlapped with site 3 (Fig 2A and 2C). In fact, mAb AM28 spanned two adjacent asymmetric units of HPeV1, with one portion binding to the knob and the other to the threefold vertex of the adjacent unit; this might stabilize the viral capsid and inhibit virus uncoating. The RGD motif was located at site 2 in HPeV1, which can bind to the receptors αvβ6 and αvβ3, and was also the linear epitope for the mAb AM18 (Fig 2A and 2C). Several residues preceding the RGD motif were generated by homology modeling. Thus, the modeled residues (VP1-216T, -217S, -218S, and -219R) were predicted as conformational epitopes, indicating that the RGD motif is both a linear and conformational epitope. Therefore, mAb AM18 can block HPeV1 infection by competitively binding site 2 with the integrin receptor.

MAb AT12-015 mainly bound to the VP1 BC loop, EF loop, C-terminus and the VP3 BC, EF and GH loops of HPeV3, spanning three sites. The VP1-85N and -87D in site 1, VP1-217G and -218S in site 2 and knob in site 3 were similar to three anchors fixing the mAb AT12-015 in the flat canyon region (Fig 2B and 2D). The neutralizing effect of mAb AT12-015 might directly or indirectly block the receptors from binding to HPeV3. Unfortunately, information on HPeV3 receptors is still unknown. Mutations of four residues, VP1-N84D, -S186L, -V195A and VP3-H206Y, can help HPeV3 evade the neutralization of mAb AT12-015 [32]. The first

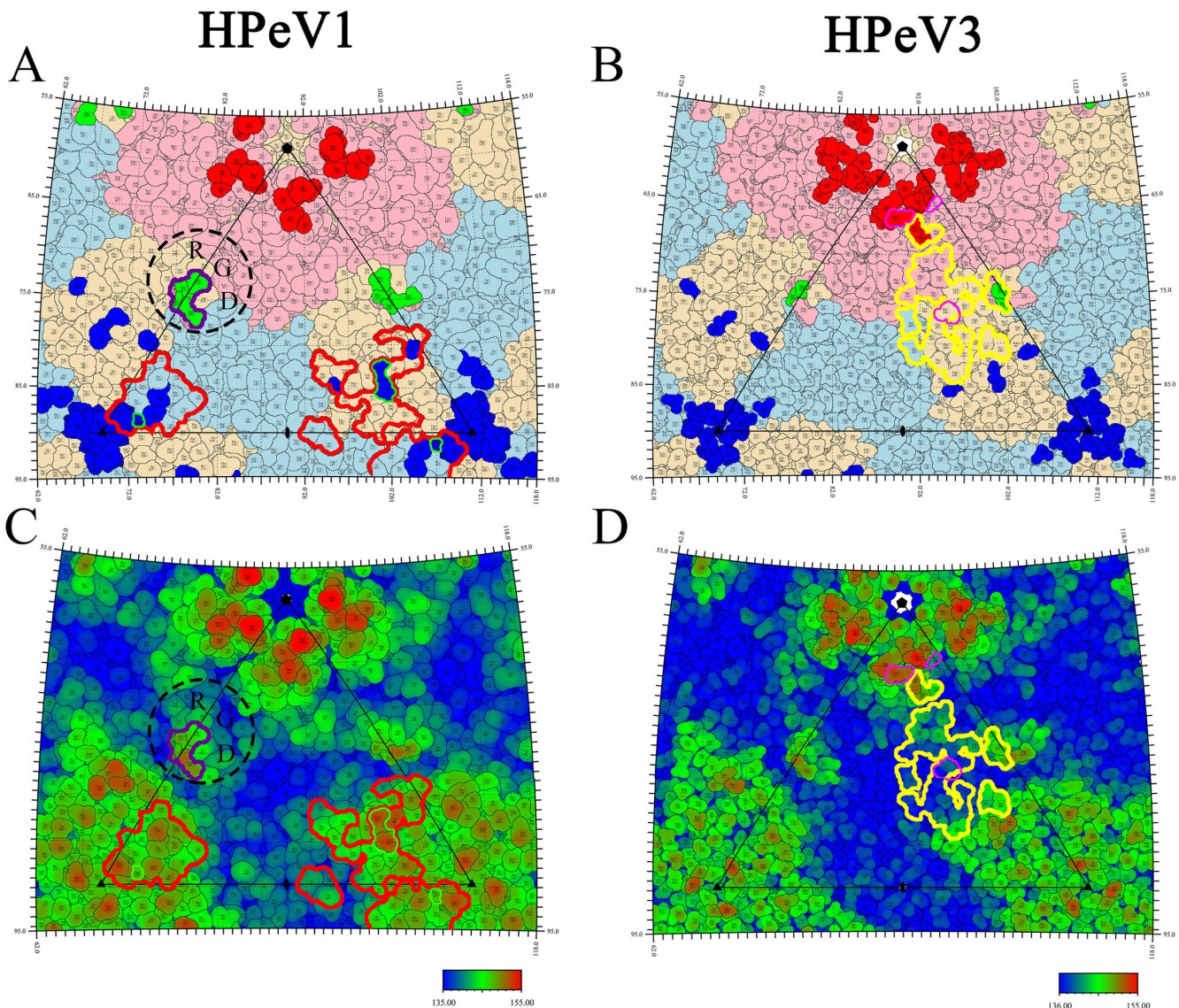

**Fig 2. Conformational epitopes and antibody footprints on the HPeV1 and HPeV3 capsid surface.** The figure shows stereographic projections of the HPeV1 and HPeV3 virion capsid surface. (A and B) Residues of VP1, VP0 and VP3 of HPeV1 (A) and HPeV3 (B) are colored in light pink, light blue and light orange, respectively; residues involved in conformational epitope sites 1, 2 and 3 are shown in red, green and blue, respectively. (A) The integrin-binding site in HPeV1 is shown as a dashed line. Regions involved in binding mAbs AM28 and AM18 are shown as a solid line colored in red and purple, respectively. The residues of VP3-91N and -92H and VP0-257S in HPeV1 are indicated by a solid green line. (B) Regions involved in binding mAb AT12-015 are shown as a solid yellow line. The residues of VP1-N84D, VP1-S186L and VP3-H206Y are shown by a solid magenta line. (C and D) The surfaces of HPeV1 (C) and HPeV3 (D) are colored blue to red, according to the distance from the particle center (red representing the furthest distance). The icosahedral asymmetric unit is also indicated by a black triangle. The positions of the icosahedral symmetry axes are indicated by pentagons (fivefold vertex), triangles (threefold vertex), and ovals (twofold vertex).

three mutations were located at site 1 (Fig 2B and 2D). VP1-84N, adjacent to VP1-85N and -87D, which is the anchor of mAb AT12-015, was in the prediction epitope. VP1-S186L and -V195A were located on the VP1 HI loop and βI sheet of HPeV3, respectively, and were partially or completely covered by site 1. Mutations in these three residues might loosen the anchoring effect of VP1-85N and -87D for mAb AT12-015.

## Comparison of conformational epitopes between HPeV1 and HPeV3

The numbers of residues contained in the VPs of HPeV1 and HPeV3 are approximately 776 and 774, respectively (S1 Fig). The VP1 of HPeV1 and HPeV3 is composed of 234 and 229 residues, respectively; VP0 contains 289 residues; and VP3 consists of 253 and 256 residues, respectively (13, 14, 16). The sequence identities between HPeV1 and HPeV3 are 71.8% (VP1), 74.9% (VP0) and 77.7% (VP3), respectively. The RMSD between HPeV1 and HPeV3 was 0.476 Å and the RMSDs of VP1, VP0 and VP3 were 0.484 Å, 0.427 Å and 0.478 Å, respectively.

The conformational epitopes of HPeV1 and HPeV3 were composed of three sites with 23 and 24 residues, respectively, of which 20 residues were shared by both serotypes. Therefore, HPeV1 and HPeV3 had highly consistent epitope distribution patterns (Table 1 and Fig 1).

At site 1, the epitopes on the VP1 BC loop of HPeV3 were four conserved residues, VP1-82D, -84N, -85N and -87D, and HPeV1 were VP1-82T, -84T, -85N/D and -87G; the epitopes on the VP1 HI loop of HPeV3 were six conserved residues, VP1-187M, -188H, -189G, -190T, -191T and -192R, and HPeV1 were VP1-187L, -188T, -189G, -190T, -191S/T and -192T (Table 1 and Fig 3). This indicated that the positions of the epitopes in site 1 of HPeV1 and HPeV3 were highly consistent, but the residues varied largely.

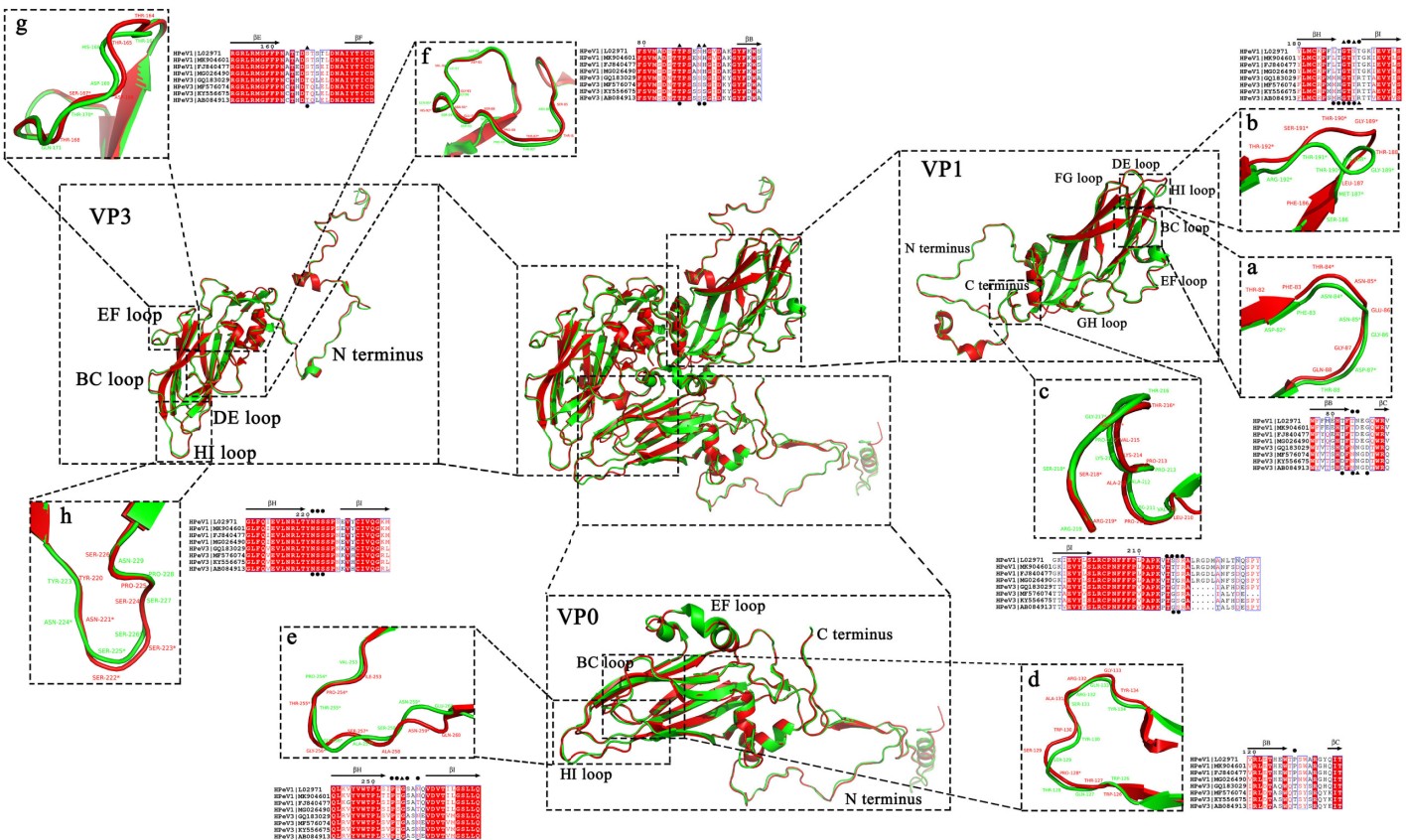

**Fig 3. Comparison of the three-dimensional structures of the conformational epitopes of HPeV1 and HPeV3.** The three-dimensional (3D) structure alignment and the corresponding close-up views of HPeV1 and HPeV3. HPeV1 and HPeV3 are colored in red and green, respectively. The sequence alignment consists of 4 strains of HPeV1 and 4 strains of HPeV3. It is adjacent to the corresponding 3D structures. (a-c) The 3D structure and sequence alignment of the BC loop (a), HI loop (b) and C-terminus (c) of VP1. (d-e) The 3D structure and sequence alignment of the BC loop (d) and HI loop (e) of VP0. (f-h) The 3D structure and sequence alignment of the N-terminus (f), EF loop (g) and HI loop (h) of VP3. A symbol (*) in the 3D structure notes that this residue is predicted as a conformational epitope. A symbol (•) or (▲) in the sequence alignment notes that this residue is predicted as a core epitope (•) or surrounding epitope (▲).

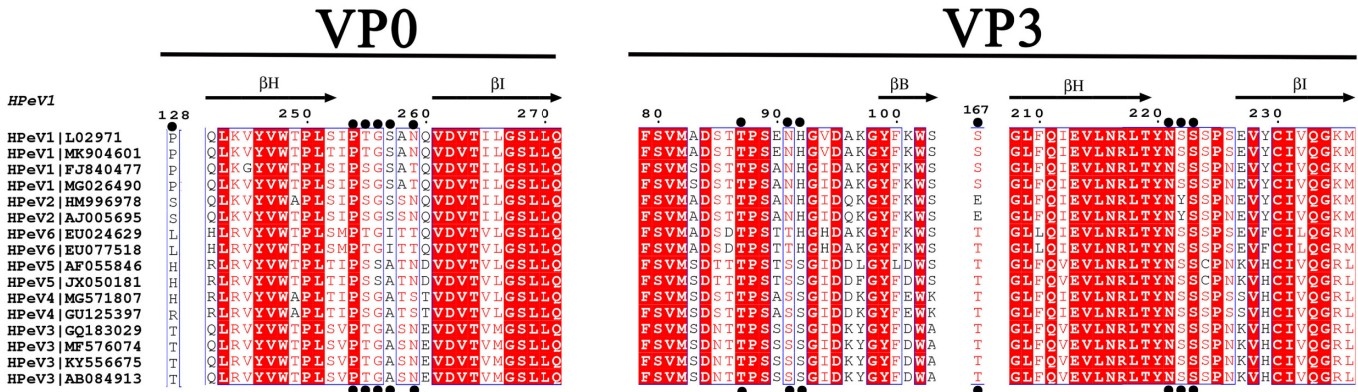

**Fig 4. Sequence alignment of the conformational epitope in site 3 among HPeV1-6.** The secondary structure elements from HPeV1 (Harris strain) are shown above the alignment as β-sheets (arrows). The predicted epitopes (black circles) for HPeV1 and HPeV3 are indicated on the corresponding protein sequence. Sequence annotations on the left side represent the virus genotypes and GenBank IDs. Sequence identities among all the HPeVs are colored on a scale of white (no identity) to red (full identity).

The RGD motif binding to the integrin receptor was located at site 2 of HPeV1 but was absent in HPeV3. It is still unclear whether this region is the binding site of the receptor in HPeV3 (Table 1 and Fig 3).

The amino acid residues in site 3 were highly conserved between HPeV1 and HPeV3, except for individual residues such as VP3-91N/94S and -92H/95S and VP0-257S/A (Table 1, Figs 3 and 4). Furthermore, multiple sequence alignments of HPeV1-19 amino acid sequences revealed that VP3-91N and -92H and VP0-257S were specific and conserved residues in HPeV1 and HPeV2 (Fig 4 and S2 Table). Intriguingly, mAb AM28 can only neutralize HPeV1 and HPeV2 but not HPeV3-6 [8], and these three residues were just the binding site of AM28 (Fig 2A and 2C). The findings mentioned above implied that VP3-91N and -92H and VP0-257S were the key sites for AM28 binding to HPeV1 and were also of great significance in distinguishing HPeV1 and HPeV3.

## Discussion

Traditionally, conformational epitopes were predicted by classical bioinformatics algorithms, including Ellipro, DiscoTope and Epitopia, which are based on the 3D structure of a protein [26–28]. Ellipro [26] represents the protein structure as a simple ellipsoid, from which the regions protruding most are more likely to be the epitopes. Therefore, calculating the protrusion index for each residue is implemented to predict the conformational epitopes. DiscoTope [27] defines a novel spatial neighborhood concept, which is the basis for employing a surface measure consisting of propensity scores and half-sphere exposure. Epitopia [28] predicts epitopes by calculating immunogenic scores that integrate features such as solvent accessibility and residue composition by the Naïve Bayes classifier. Borley *et al.* [24] used the consistent results of three prediction tools (Ellipro, DiscoTope and Epitopia) as the final predicted epitopes, which facilitated the specificity and accuracy of the prediction results. More importantly, traditional algorithms only treat a VP as an isolated protein when predicting epitopes. Actually, in picornaviruses, such as FMDV and HPeV, the VPs are not isolated but constitute an asymmetric unit together. Then, 60 asymmetric units are assembled into a capsid. In this circumstance, parts of the VPs are buried in the capsid and unlikely to be conformational epitopes (exceptionally, there are some epitopes underneath the capsid that have contact with the outside of the capsid through "capsid breathing" [10, 33]). Therefore, residues buried in the

capsid that are successfully predicted as epitopes can be considered false positives of algorithms. To decrease false positives and vastly improve the specificity and accuracy of prediction results, Borley *et al.* built multiple chains to mimic the whole capsid for predictions (if the complete capsid is used, this would far exceed the computational power of the existing algorithm). Thus, this algorithm has broad application prospects in predicting conformational epitopes of picornaviruses.

Since the resolutions of 3D structures are not completely the same and some random errors occur in structural determination, we have made some modifications for Borley *et al.*'s algorithm by defining the concepts of "core epitope" and "surrounding epitope" based on the assumption "similar structure, similar epitope". The final prediction results of epitopes consist of core epitopes and surrounding epitopes. Core epitope refers to the consistent results among all three prediction tools as Borley *et al.* did. Surrounding epitope need the assistance of another serotype/serotypes with the evolutionary close relationship such as the serotype of the same species. For example, VP3 knob of HPeV1 (-87T, -91N and -92H) was predicted as epitope only by two tools, however, it was the core epitope of HPeV3 (S1 Fig). Thus, it was considered as the surrounding epitope of HPeV1. Actually, VP3 knob is an important epitope of HPeV1: mAb AM28 binds to the regions including VP3 knob; -91N and -92H may be specific residues for the binding of mAb AM28 to HPeV1 and HPeV2 (see results). This showed that our modification may avoid the Borley *et al.*'s excessively rigid algorithm. In addition, some residues on the surface of the capsid (such as the VP1 C-terminus) were missing in the 3D structure, which not only directly affected the prediction, but also exposed the internal residues. This may lead to false positive prediction results. In fact, epitopes were usually located on highly flexible loops where the deletions were more likely to occur. Therefore, the missing residues were imputed through homology modeling. Furthermore, because the HPeVs had long C-terminus, it was impossible to directly generate multiple chains using Borley *et al.*'s method. A complete capsid was constructed and then was used to generate multiple chains.

In a previous study, we applied this method to systematically predict the conformational epitopes of CVA10 [17] and found that these epitopes clustered into three sites (sites 1, 2 and 3): site 1 is located on the northern rim of the canyon near the fivefold vertex, and site 2 is on the puff. Site 3 is divided into two parts: one is located on the knob, and the other is close to the threefold vertex. This epitope distribution pattern was highly consistent with those from the other two human enteroviruses, rhinovirus B14 (RVB14) and PV1, suggesting that it is the basic pattern of epitope distribution in human enteroviruses [34, 35]. In the present study, we explored whether this epitope distribution pattern exists on HPeV. We found that the conformational epitopes of HPeV1 and HPeV3 also present a three-cluster distribution pattern, which is basically consistent with enteroviruses such as CVA10. Compared to CVA10, fewer residues are contained on the VP0 EF loop and VP1 GH loop. The puff is flat and it is difficult to form conformational epitopes, for which the site 2 in HPeV1 and HPeV3 is mainly composed of the VP1 C-terminus. The number of residues predicted to be epitopes is much less in HPeV1 (23 residues) and HPeV3 (24 residues) than that in CVA10 (45 residues), which is consistent with the flatter surfaces of the HPeV1 and HPeV3 capsids.

The distribution patterns of conformational epitopes in HPeV1 and HPeV3 are highly consistent, but the residues that form the epitopes are different. And the difference degree is similar to that of the human enterovirus A71 (EV71) and coxsackievirus A16 (CVA16) (unpublished data). At site 1, there are a total of 10 residues in the HPeV3 epitopes, half of which differ from those in HPeV1. The RGD motif was located at site 2 in HPeV1, which can bind to the receptors αvβ6 and αvβ3, and it was also the linear epitope for the mAb AM18. Therefore, mAb AM18 can block HPeV1 infection by competitively binding site 2 with the integrin receptor. Similar to HPeV1, FMDV uses RGD motif (VP1 GH loop instead of VP1 C-

terminus) to bind integrin receptors and it could be blocked by mAbs like mAb 10D5 [36]. However, integrin is only an attachment receptor for HPeV1 and there may be an unidentified receptor involved for the capsid uncoating, which binds to the predicted epitopes in the study. Site 2 in HPeV3 lacks the RGD motif possessed by HPeV1 and fails to bind to integrin receptors [9]. Although the residues on site 3 in both viruses are highly homogeneous, the variations of the three key residues VP3-91N/94S, VP3-92H/95S and VP0-257S/A have an impact on the binding capacity of the mAb AM28. This demonstrates that the mutations of these residues in the epitopes generate different HPeV genotypes. In HPeV1, the binding sites of the integrin receptor and mAb AM18 are located in site 2, and the site of mAb AM28 is located in site 3 [8]. However, there is no report showing that mAb can bind to site 1. Site 1 is located in the northern rim of the canyon, which is also the location of epitope NIm-I of RVB14 [34], N-AgI of PV1 [35] and mAb 28–7 binding to EV71 [37]. These results suggest that site 1 has a high probability of being a conformational epitope. Only one mAb of HPeV3 [7, 9] has been reported, named AT12-015, and the mode of this mAb binding to the viral capsid is very special. It not only binds to the relatively protruding residues on the capsid surface but also crosses three sites to bind to the flat canyon (Fig 2B and 2D). Strain-dependent neutralization also proved that the "anchor" VP1-85N and -87D on site 1 is very important for the binding of AT12-015 [32].

In summary, we developed a bioinformatics algorithm based on Borley *et al.*'s multiple chains to predict the conformational epitopes for HPeVs and found that the epitopes of HPeV1 and HPeV3 also clustered into three sites. HPeV1 and HPeV3 have a highly consistent epitope distribution pattern, but the mutant residues at the epitopes produce different HPeV genotypes. We hope that the exploration of epitopes of HPeVs can help monitor their antigenic evolution like influenza viruses [12–16]. Additionally, the three residues (VP3-91N and -92H and VP0-257S) on site 3 may play a pivotal role in AM28 binding to HPeV1, and two residues (VP1-85N and -87D) on site 1 may be the key residues of AT12-015 binding to HPeV3.

## Conclusions

We found that the conformational epitopes of HPeV1 and HPeV3 had similar distribution patterns. Five residues involved in conformational epitopes were key sites for mAb binding. This study is of great significance to understand the pathogenesis mechanisms of HPeV infections and develop antiviral antibodies.

## Supporting information

**S1 Table. Comparison of our and Borley et al.'s prediction results for FMDV.** (DOCX)

**S2 Table. The sequence conservation of VP3-91, -92 and VP0-257 among HPeV1-19.** (DOCX)

**S1 Fig. The sequence alignment of VP1, VP0 and VP3 of HPeV1 and HPeV3.** The secondary structure elements and predicted epitopes from the HPeV1and HPeV3 are shown on the corresponding protein sequence alignment as β-sheets (arrows), α-helices (spirals), and the predicted core epitopes (black circles) and surrounding epitopes (black triangles). Sequence annotations on the left side represent to the virus genotypes and corresponding VP. Sequence identities between HPeV1 and HPeV3 are colored on a scale of white (no identity) to red (full identity). (DOCX)

**S2 Fig. The schematic depiction of core epitopes and surrounding epitopes.** The symbols used in this figure are the same as S1 Fig.
(DOCX)

**S3 Fig. Predicted conformational epitopes of the multiple chains of HPeV1 and HPeV3.** The complete conformational epitopes of the multiple chains of HPeV1 (A) and HPeV3 (B) are presented. The colors and symbols used in this figure are the same as those in Fig 1.
(DOCX)

## Author Contributions

**Conceptualization:** Hao Rong, Jianli Hu, Changzheng Dong.

**Data curation:** Hao Rong, Changzheng Dong.

**Formal analysis:** Hao Rong, Liping Wang, Liuying Gao, Yulu Fang, Qin Chen, Jianli Hu, Lina Zhang.

**Funding acquisition:** Changzheng Dong.

**Methodology:** Hao Rong.

**Project administration:** Changzheng Dong.

**Supervision:** Changzheng Dong.

**Validation:** Changzheng Dong.

**Writing – original draft:** Hao Rong, Liping Wang, Yulu Fang, Qin Chen, Meng Ye, Qi Liao, Lina Zhang, Changzheng Dong.

**Writing – review & editing:** Hao Rong, Jianli Hu, Changzheng Dong.

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
