## [Decision Letter · Decision Letter 0]

24 Nov 2020

PONE-D-20-27668

Bioinformatics-based prediction of conformational epitopes for human parechovirus

PLOS ONE

Dear Dr. Dong,

Thank you for submitting your manuscript to PLOS ONE. After careful consideration, we feel that it has merit but does not fully meet PLOS ONE’s publication criteria as it currently stands. Therefore, we invite you to submit a revised version of the manuscript that addresses the points raised during the review process.

Both reviewers noticed that the main method used for the study was adapted from the original publication but that the modifications were not completely described. This is not in line with the PLOS publication criteria and must be addressed.

The manuscript should also be adapted to be more clear wherever possible and include all relevant introduction, method description and discussion as indicated by the reviewers.

The issues to address are listed below

We look forward to receiving your revised manuscript.

Kind regards,

Ivan Sabol

Academic Editor

PLOS ONE

Journal Requirements:

Reviewers' comments:

Reviewer's Responses to Questions

**Comments to the Author**

1. Is the manuscript technically sound, and do the data support the conclusions?

Reviewer #1: Partly

Reviewer #2: Yes

2. Has the statistical analysis been performed appropriately and rigorously? 

Reviewer #1: N/A

Reviewer #2: N/A

3. Have the authors made all data underlying the findings in their manuscript fully available?

Reviewer #1: Yes

Reviewer #2: Yes

4. Is the manuscript presented in an intelligible fashion and written in standard English?

Reviewer #1: No

Reviewer #2: Yes

5. Review Comments to the Author

Reviewer #1: In the manuscript entitled ‘Bioinformatics-based prediction of conformational epitopes for Human Parechovirus’ Rong et al used previously published bioinformatics tools (Borley et al, 2013) with some modifications to predict conformational epitopes on the surface of parechovirus capsids. This study might be of interest for therapeutic antibody development mainly to treat neonatal parechovirus infections.

The manuscript in its current form, however, is not suitable for publication.

Major points:

The introduction is not comprehensively written. The research question and the related topic is not properly introduced and it is very hard for the reader to grasp what has been done and why. The method and why it was chosen is not clear. In addition, it is not clear why authors chose parechoviruses for their analysis. They mention that it is important for vaccine design and monitoring of antigenic evolution like in case with influenza. Parechoviruses are not comparable with influenza virus because (as authors mentioned) seroprevalence increases with age and 6 year and older children have 100% protection for HPeV1 and 85% for HPeV3. Parechoviruses are the main threat for neonates for whom vaccines is not an option. Authors should reconsider the impact of their study.

It is very confusing why authors add published results obtained with CVA10.

The accuracy of conformational epitope prediction largely depends on the accuracy of 3D structures of viruses of interest. It is not mentioned in the text how accurate structures for HPeV1 and HPeV3 are, which were basis for current study. Authors also mention that part of the polypeptide missing from the structures was modelled which hampers the accuracy of the structure. In addition, there are very few experimentally confirmed epitopes to validate the results. Authors may argue that the method has been published, but there were some modifications introduced.

Methods are not sufficiently described.

The result section sounds very repetitive. Additionally, for comparison authors randomly chose only few isolates from 6 parechovirus types. It is of importance to evaluate the similarity of predicted epitopes in as large number of isolates as possible from all parechovirus types to make some conclusions.

Part of Figure 1 and Suppl Figures 1-3 show only trivial information published elsewhere in a very similar way. Should be excluded. Authors should make less repetitive figures (eg Suppl Fig 1 and 2) clearly showing the point of interest.

Specific points:

Page 3 lines 55-57: it is not clear why authors choose CVA10 for comparison, as this is relevant to all picornaviruses.

Page 3 line 59: term ‘viral maturation’ might be misleading in this context. Should be rephrased.

Page 4 line 65: It is very obscure what is ‘structural protein’? It should be ‘part of polyprotein encoding for structural proteins’.

Page 4 line 69: it is not clear how many strains are included in the comparison.

70-71: it is true for all picornaviruses thus it is misleading to mention only enteroviruses.

Page 4 lines 73-78: Authors describe data published by others (Shakeel et al 2016, Domanska et al, 2019, and Kalynych et al, 2015), thus the Fig 1A-F has no additional value.

Page 5 line 93: This is definition for conformational epitopes. It is unclear what authors mean by ‘B-cell conformational epitopes’. Needs further explanation.

Table 1. It is not clear why authors add already published results into the Table 1 alongside with unpublished data. The better place for published data is discussion.

Reviewer #2: In this paper, Rong et al presented analysis of conformational epitopes of HPeV1 and HPeV3 using a modified protocol based on the work of Borley et al. They identified three potential sites of conformational epitopes, of which two can be corroborated with previous work. All three sites have also been identified in CVA10 in the author’s previous work. Overall, the work is well presented and of interest within the picornavirus community.

I have few comments on the manuscripts:

1. Please expand more how you modified the method described in Borley et al.

2. Can the authors re-do the analysis of FMDV which formed the basis of the work of Borley et al to see whether their modification in the protocol affected the analysis of FMDV as well. And also include this analysis on FMDV as the second comparison besides the one they do with CVA10.

3. Can the authors comment on whether the receptor-binding sites overlap with the conformational epitopes for other picornaviruses for which receptors as well as conformational epitopes are known? They can include a comment on HPeV1 that even though the conformational epitope of HPeV1 do not overlap with the integrin-binding site, but the integrin is only an attachment receptor and there may be, yet another unidentified receptor involved for destablizing the capsid for genome release which may be using the same site as the conformational epitopes identified in this study.

4. Can the authors include a Q-score from PDBeFold or some other structural similarity measure such as rmsd between the HPeV1 and HPeV3 capsid structures. This would provide an idea about overall structural similarity between these viruses.

5. In line 35, please add monoclonal antibody in front of AM28.

6. In line 151, please remove the word “crystal” as there is no crystal structure of HPeV3 VP1.

7. In line 280, please replace * with a black dot because that’s what you in the corresponding figure 3.

8. In line 350, please clarify what the authors meant by “Dissimilarly”.

6. PLOS authors have the option to publish the peer review history of their article (what does this mean?). If published, this will include your full peer review and any attached files.

Reviewer #1: No

Reviewer #2: No

---

## [Author Response · Author response to Decision Letter 0]

20 Jan 2021

To reviewers:

Thanks for the careful reviewing and constructive suggestions. These suggestions improve the quality of the manuscript greatly. In this study, we modified Borley’s algorithm and applied to the prediction of the conformational epitopes for HPeV1 and HPeV3. In the revision, we highlighted the modification and compared the prediction results for foot and mouth disease virus (FMDV) with Borley et al.’s. The comparison showed that our algorithm can capture more epitopes than Borley et al.’s and exhibited more clear distribution patterns of epitopes. Since HPeVs and coxsackievirus A10 (CVA10) both belong to the Picornaviridae family and have analogous capsid surface structures, we compared HPeVs’ structures and epitopes with CVA10 (our published work) previously. In the revision, we removed this part and just discussed briefly in the discussion.

Thanks again.

Yours sincerely,

Dr. Changzheng Dong

Ningbo University School of Medicine

 

Reviewer #1: In the manuscript entitled ‘Bioinformatics-based prediction of conformational epitopes for Human Parechovirus’ Rong et al used previously published bioinformatics tools (Borley et al, 2013) with some modifications to predict conformational epitopes on the surface of parechovirus capsids. This study might be of interest for therapeutic antibody development mainly to treat neonatal parechovirus infections. The manuscript in its current form, however, is not suitable for publication.

Authors’ response: Thanks for the comment. We revised the manuscript intensely and hope it could meet the journal’s publication criteria.

Major points: The introduction is not comprehensively written. The research question and the related topic is not properly introduced and it is very hard for the reader to grasp what has been done and why. The method and why it was chosen is not clear. In addition, it is not clear why authors chose parechoviruses for their analysis. They mention that it is important for vaccine design and monitoring of antigenic evolution like in case with influenza. Parechoviruses are not comparable with influenza virus because (as authors mentioned) seroprevalence increases with age and 6 year and older children have 100% protection for HPeV1 and 85% for HPeV3. Parechoviruses are the main threat for neonates for whom vaccines is not an option. Authors should reconsider the impact of their study.

Authors’ response: Thanks for the suggestions. The abstract and the introduction have been rewritten. “Human parechoviruses (HPeVs) are human pathogens that usually cause diseases ranging from rash to neonatal sepsis in young children. HPeV1 and HPeV3 are the most frequently reported genotypes and their three-dimensional structures have been determined. However, there is a lack of systematic research on the antigenic epitopes of HPeVs, which are useful for understanding virus-receptor interactions, developing antiviral agents or molecular diagnostic tools, and monitoring antigenic evolution. Thus, we systematically predicted and compared the conformational epitopes of HPeV1 and HPeV3 using bioinformatics methods in the study.” Borley et al.’s algorithm is a good method for the picornaviruses’ epitope prediction. We modified the algorithm and applied to HPeVs. 

It is very confusing why authors add published results obtained with CVA10.

Authors’ response: Thanks for the suggestion. Since the conformational epitopes of CVA10 has been predicted using the similar method, we wanted to compare HPeVs with CVA10 previously. In the revision, the CVA10 related contents are deleted from the introduction, methods and results. But a brief discussion is kept in the discussion. Now the manuscript seems more concise. 

The accuracy of conformational epitope prediction largely depends on the accuracy of 3D structures of viruses of interest. It is not mentioned in the text how accurate structures for HPeV1 and HPeV3 are, which were basis for current study. 

Authors’ response: Thanks for the suggestion. The atomic-level resolutions for HPeV1 (4Z92) and HPeV3 (6GV4) are 3.10 Å and 2.80 Å, respectively. It is provided in the revision.

Authors also mention that part of the polypeptide missing from the structures was modelled which hampers the accuracy of the structure. 

Authors’ response: Several residues on the VP1 C-terminus (including the RGD motif) were missing in the 3D structure of HPeV1. The phenomena is common for picornaviruses due to the flexibility of the loops and termini, which is the characteristic of epitopes. We modeled only three residues (VP1-217S, -218S and -219R) in HPeV1 based on the conserved residues from HPeV3 without missing in the 3D structure. The predicted epitope based on the homology modeling is also marked (Table 1). 

In addition, there are very few experimentally confirmed epitopes to validate the results. Authors may argue that the method has been published, but there were some modifications introduced.

Authors’ response: Thanks for the suggestion. The accuracy of the algorithm is assessed with various evidences. Firstly, the algorithm is compared to Borley et al.’s according to the results of FMDV (provided in the revision). Secondly, the direct evidences from the experimentally confirmed epitopes. Thirdly, the indirect evidences from CVA10 and other enteroviruses such as rhinovirus and poliovirus. 

Methods are not sufficiently described.

Authors’ response: We revised the methods especially for the algorithm modification.

The result section sounds very repetitive. Additionally, for comparison authors randomly chose only few isolates from 6 parechovirus types. It is of importance to evaluate the similarity of predicted epitopes in as large number of isolates as possible from all parechovirus types to make some conclusions.

Authors’ response: Thanks for the suggestion. In the revision, we used all HPeV1-6 strains with the complete genome sequences from NCBI Nucleotide database (S2 Table) and the results are same. For the aim of illustration, Figs 3 and 4 use limited strains as before. 

Part of Figure 1 and S1-S3 Fig show only trivial information published elsewhere in a very similar way. Should be excluded. Authors should make less repetitive figures (eg S1 and S2 Fig) clearly showing the point of interest.

Authors’ response: Thanks for the suggestion. In the revision, the CVA10 related contents are deleted (previous S1 and S2 Fig) and Fig 1 was also modified. Now the manuscript seems more concise. 

Specific points:

Page 3 lines 55-57: it is not clear why authors choose CVA10 for comparison, as this is relevant to all picornaviruses.

Authors’ response: Since the conformational epitopes of CVA10 has been predicted using the similar method, we wanted to compare HPeV with CVA10 previously. Now the CVA10 related contents are deleted from the introduction, methods and results. But a brief description is kept in the discussion. 

Page 3 line 59: term ‘viral maturation’ might be misleading in this context. Should be rephrased.

Authors’ response: The term “viral maturation” was deleted in the revision. 

Page 4 line 65: It is very obscure what is ‘structural protein’? It should be ‘part of polyprotein encoding for structural proteins’.

Authors’ response: We used “viral protein (VP)” instead of “structural protein” in the revision. 

Page 4 line 69: it is not clear how many strains are included in the comparison.

Authors’ response: In the paragraph (Fig S1), only two strains are used to estimate the sequence identity between HPeV1 and HPeV3. Since they are two different genotypes (serotypes), the identity varies slightly if more strains are included. 

70-71: it is true for all picornaviruses thus it is misleading to mention only enteroviruses.

Authors’ response: Previously we aimed to compare with enteroviruses such as CVA10. Now the related contents are deleted in the revision. 

Page 4 lines 73-78: Authors describe data published by others (Shakeel et al 2016, Domanska et al, 2019, and Kalynych et al, 2015), thus the Fig 1A-F has no additional value.

Authors’ response: Fig 1 exhibits the multiple chains and the predicted epitopes on the capsids to help understand the algorithm and the distribution patterns of the epitopes. However, it did not use any published data by others. Actually, those data (Shakeel et al, 2016, and Domanska et al, 2019) were used in Fig 2 to assess the accuracy of the prediction. 

Page 5 line 93: This is definition for conformational epitopes. It is unclear what authors mean by ‘B-cell conformational epitopes’. Needs further explanation.

Authors’ response: we used “conformational epitopes” directly instead of ambiguous word “B-cell conformational epitopes” in the revision.

Table 1. It is not clear why authors add already published results into the Table 1 alongside with unpublished data. The better place for published data is discussion.

Authors’ response: we deleted the content about comparing with CVA10 from the introduction, methods and results. 

Reviewer #2: In this paper, Rong et al presented analysis of conformational epitopes of HPeV1 and HPeV3 using a modified protocol based on the work of Borley et al. They identified three potential sites of conformational epitopes, of which two can be corroborated with previous work. All three sites have also been identified in CVA10 in the author’s previous work. Overall, the work is well presented and of interest within the picornavirus community.

I have few comments on the manuscripts:

1. Please expand more how you modified the method described in Borley et al.

Authors’ response: Thanks for the suggestion. We highlighted the modification and compared to Borley et al.’s algorithm in the methods, results and discussion. 

2. Can the authors re-do the analysis of FMDV which formed the basis of the work of Borley et al to see whether their modification in the protocol affected the analysis of FMDV as well. And also include this analysis on FMDV as the second comparison besides the one they do with CVA10.

Authors’ response: Thanks for the suggestions. It’s a good idea to assess the accuracy of our modification. We reanalyzed the epitopes of FMDV and compared to Borley et al.’s. We revised the manuscript in the methods, results and discussion.

3. Can the authors comment on whether the receptor-binding sites overlap with the conformational epitopes for other picornaviruses for which receptors as well as conformational epitopes are known? They can include a comment on HPeV1 that even though the conformational epitope of HPeV1 do not overlap with the integrin-binding site, but the integrin is only an attachment receptor and there may be, yet another unidentified receptor involved for destablizing the capsid for genome release which may be using the same site as the conformational epitopes identified in this study.

Authors’ response: Thanks for the suggestions. Similar to HPeV1, FMDV uses RGD motif (VP1 GH loop instead of VP1 C-terminus) to bind integrin receptors and it could be blocked by mAbs. For example, the binding of integrin αvβ6 can be specially inhibited by mAb 10D5 (Terry Jackson et al., J Virol 2000). Just like the reviewer’s comment, the unidentified uncoating receptor for HPeV1 may bind to the predicted conformational epitopes in the study. We provide these comments in the discussion.

4. Can the authors include a Q-score from PDBeFold or some other structural similarity measure such as rmsd between the HPeV1 and HPeV3 capsid structures. This would provide an idea about overall structural similarity between these viruses.

Authors’ response: Thanks for the suggestion. The root mean square deviation (RMSD) of structural difference (Cα atoms in a asymmetric unit) calculated by PyMol between HPeV1 and HPeV3 is 0.476 Å and the RMSDs of VP1, VP0 and VP3 are 0.484 Å, 0.427 Å and 0.478 Å, respectively, which exhibits a high structural similarity between these two genotypes. It was provided in the revision. 

5. In line 35, please add monoclonal antibody in front of AM28.

Authors’ response: It was corrected in the revision.

6. In line 151, please remove the word “crystal” as there is no crystal structure of HPeV3 VP1.

Authors’ response: In the sentence, “crystal structure” should be “3D structure”. It was corrected in the revision.

7. In line 280, please replace * with a black dot because that’s what you in the corresponding figure 3.

Authors’ response: There are both symbols “*” and black dot in the figure 3 and we forgot to note the meaning of the symbol “*”. It was corrected in the revision.

8. In line 350, please clarify what the authors meant by “Dissimilarly”.

Authors’ response: The ambiguous word “Dissimilarly” was deleted. 

 

Major revision list:

1. The abstract and the introduction have been rewritten.

2. Table 1 and Fig 1 were modified to remove CVA10 related. Previous S2 and S3 Figs (CVA10 related) were deleted. New S2 Fig was provided to present core and surrounding epitopes. S1 Table was provided to compare the algorithm with Borley et al.’s. S2 Table was used to describe the sequence conservation among HPeV1-6. 

3. CVA10 related contents were deleted from introduction, materials and methods, and results. 

4. Algorithm modification to Borley et al.’s was highlighted in materials and methods, results, and discussion. The epitopes of FMDV were predicted and compared them with Borley's results and known epitopes to assess the algorithm accuracy.

---

## [Decision Letter · Decision Letter 1]

8 Feb 2021

Bioinformatics-based prediction of conformational epitopes for human parechovirus

PONE-D-20-27668R1

Dear Dr. Dong,

We’re pleased to inform you that your manuscript has been judged scientifically suitable for publication and will be formally accepted for publication once it meets all outstanding technical requirements.

Reviewer 1 has made some small suggestions that could further improve the manuscript if possible. These changes will not go through another round of peer review.

Kind regards,

Ivan Sabol

Academic Editor

PLOS ONE

Additional Editor Comments (optional):

Reviewers' comments:

Reviewer's Responses to Questions

**Comments to the Author**

1. If the authors have adequately addressed your comments raised in a previous round of review and you feel that this manuscript is now acceptable for publication, you may indicate that here to bypass the “Comments to the Author” section, enter your conflict of interest statement in the “Confidential to Editor” section, and submit your "Accept" recommendation.

Reviewer #1: (No Response)

Reviewer #2: All comments have been addressed

2. Is the manuscript technically sound, and do the data support the conclusions?

Reviewer #1: Yes

Reviewer #2: Yes

3. Has the statistical analysis been performed appropriately and rigorously? 

Reviewer #1: I Don't Know

Reviewer #2: N/A

4. Have the authors made all data underlying the findings in their manuscript fully available?

Reviewer #1: Yes

Reviewer #2: Yes

5. Is the manuscript presented in an intelligible fashion and written in standard English?

Reviewer #1: Yes

Reviewer #2: Yes

6. Review Comments to the Author

Reviewer #1: The revised manuscript by Rong et al entitled ‘Bioinformatics-based prediction of conformational epitopes for Human Parechovirus’ has greatly improved compared to previous version. Authors described modification of the algorithm originally used by Borley et al. in more detail. Authors also verified and compared new modification using previously published data by Borley et al. The repetitive content has been removed; abstract and introduction have been revised, and discussion modified accordingly.

Could authors still include the comprehensive list of HPeV types in the sequence conservation analysis? Currently authors provide data for only HPeV1-6 types, whereas there are 19 types recognized (S2 Table).

P 3 line 60, please use ‘Like other picornaviruses’ instead of ‘Like most picornaviruses’.

Reviewer #2: (No Response)

7. PLOS authors have the option to publish the peer review history of their article (what does this mean?). If published, this will include your full peer review and any attached files.

Reviewer #1: No

Reviewer #2: No

---

## [Editor Report · Acceptance letter]

23 Mar 2021

PONE-D-20-27668R1 

Bioinformatics-based prediction of conformational epitopes for Human Parechovirus 

Dear Dr. Dong:

I'm pleased to inform you that your manuscript has been deemed suitable for publication in PLOS ONE. Congratulations! Your manuscript is now with our production department. 

Kind regards, 

on behalf of

Dr. Ivan Sabol 

Academic Editor

PLOS ONE